# Communication of an Abnormal Metabolic Newborn Screening Result in the Netherlands: A Qualitative Exploratory Study of the General Practitioner’s Perspective

**DOI:** 10.3390/ijns11030062

**Published:** 2025-08-08

**Authors:** Sietske Haitjema, Charlotte M. A. Lubout, Justine H. M. Zijlstra, Rendelien K. Verschoof-Puite, Francjan J. van Spronsen

**Affiliations:** 1Department of Metabolic Diseases, Beatrix Children’s Hospital, University Medical Center Groningen, University of Groningen, 9700 RB Groningen, The Netherlands; s.haitjema@umcg.nl (S.H.);; 2Department of Vaccine Supply and Prevention Programmes, National Institute for Public Health and the Environment (RIVM), 3720 BA Bilthoven, The Netherlands

**Keywords:** phenylketonuria, newborn screening, communication, general practitioner

## Abstract

Newborn screening (NBS) for inherited metabolic diseases (IMD) aims to find children in which immediate action can prevent severe symptoms. We previously studied parental satisfaction with the communication of the NBS result for phenylketonuria, which in the Netherlands is done by the general practitioners (GPs). More than half of all parents were unsatisfied with the communication of the abnormal NBS result. The aim of this qualitative exploratory study was to portray a number of GPs’ opinions and experiences in communicating an abnormal metabolic NBS result. We performed semi-structured interviews with ten GPs to evaluate the process of communicating the abnormal NBS result. An additional two GPs provided their answers via email. The data revealed four key themes: (1) dealing with the urgency of the metabolic NBS result, (2) the role of the GP in the NBS process, (3) the current organization of NBS in the Netherlands and (4) evaluating roles and responsibilities in communicating abnormal metabolic NBS results. Despite the willingness of GPs to inform parents about NBS results, it is questionable whether they have the necessary tools to effectively conduct these conversations given their limited experience with IMDs. In light of the increasing number of diseases in the NBS program, it would be interesting to explore alternative tools for communicating the NBS result to parents.

## 1. Introduction

After a pilot of 5 years in the northern part of the Netherlands, newborn screening (NBS) was initiated in the whole country in 1974, which initially only included phenylketonuria (PKU) [1]. Currently, this NBS program includes 27 disorders, of which 19 are an inherited metabolic disease (IMD) [2]. The way the abnormal NBS result is communicated to the parents is different in many countries [3]. In the Netherlands, a heel prick test is performed between 72 and 168 h after birth. In the case of an abnormal result, the National Institute for Public Health and the Environment (RIVM) regional office’s medical advisor contacts the general practitioner (GP), gives oral and written information about the screening result and the specific disease, and explains the actions that need to be taken. The GP is responsible for informing the family and first evaluation of the newborn with an abnormal NBS result, after which they are referred to a pediatrician for IMDs in a university medical center [4]. For most IMDs, it is of major importance that the newborn is evaluated in a metabolic center as soon as possible (within 24 h) after the result has become known.

The news of an abnormal NBS result can be very distressing for parents and has consequences on both short- and long-term outcomes in both children and parents [5,6,7]. Chudleigh et al. [8] recently reported on health professionals’ experiences in the communication of an abnormal NBS result, showing that the communication varied strongly among healthcare professionals, mainly influenced by the available resources. They developed interventions to help standardize communication of an abnormal NBS result in the United Kingdom [9]. These studies did not only focus on urgent diseases like IMDs but also on less urgent diseases screened for by NBS.

The number of IMDs screened for in NBS keeps growing. In addition, the information that needs to be provided by the GP differs between IMDs. As mentioned before, the vast majority of IMDs require an urgent referral. It is therefore difficult for the GP to be well prepared to inform the parents about the abnormal NBS result of an IMD, especially since a GP encounters such cases only once or twice during their career [6]. We previously showed dissatisfaction among parents regarding the communication of an abnormal metabolic NBS result due to the receival of wrong or incomplete information [10], among others.

To improve the communication of an abnormal metabolic NBS result to parents, it is important to obtain a clear picture of how GPs experience this process. The aim of this qualitative exploratory study is to portray a number of GPs’ experiences with the first communication of an abnormal metabolic NBS result in the Netherlands, in order to think of additional tools to improve this communication.

## 2. Materials and Methods

Qualitative, semi-structured interviews were developed in close cooperation with the National Institute for Public Health and the Environment (RIVM) and with IMD pediatricians. The semi-structured interviews were divided into three main parts: (1) the preparation of the conversation with parents, (2) the conversation itself (3) and the referral and process afterwards.

Recruitment took place through the Advice Committee Neonatal Screening for IMDs (ANS-MZ). IMD pediatricians (involved in the ANS-MZ) were asked to:Contact the GP who referred the child with an abnormal NBS result.Ask if the GP is willing to receive information about a study regarding the communication of an abnormal metabolic NBS result.Send the contact details from the GP to the researcher (SH) in case of the GP’s approval.

Next, GPs received an information letter including the proposed interview questions and an informed consent form by email. Non-responders received two follow-up invitations after one and three weeks. Respondents were asked to provide a signed informed consent form to participate in the interview. Online interviews were scheduled via Teams. Interviews were performed in Dutch. The interviews were recorded, transcribed and thereafter deleted. The study protocol was approved by the Medical Ethics Review Board of Groningen (METc number 202200335).

Data were analyzed using thematic analysis, including the following six steps: (1) familiarization with data, (2) generating initial codes, (3) searching for themes, (4) reviewing themes, (5) defining and naming themes and (6) producing reports [11]. Interviews were conducted and transcribed by one researcher (SH). The core analytical team was formed by two researchers (SH, JZ). Initial codes and themes were identified based on the first three interviews using a latent approach [11]. Interviews were continued until thematic saturation was reached, meaning that no new themes or insights emerged from the data [12]. Final coding was performed by one researcher (SH) using Atlas.ti Software (version 23) and reviewed by a second researcher (JZ). Codes and themes were discussed and reviewed in several group meetings with the research team.

## 3. Results

A total of 23 GPs agreed to receive information about this study, of whom 12 eventually participated. Non-responders did not provide reasons for not participating. Two GPs submitted written answers on the semi-structured interview via email. Online interviews were conducted with ten GPs; two GPs were involved with the child but did not communicate the NBS result themselves. Interviews ranged in duration from 12 to 23 min, with a median of 17 min.

Of the twelve included GPs, nine indicated it was their first time to communicate an abnormal NBS result to parents. Four key themes were identified after completion of the interviews: (1) dealing with the urgency of the metabolic NBS result, (2) the role of the GP in the NBS process, (3) the current standard in the Netherlands and (4) evaluating roles and responsibilities in communicating abnormal metabolic NBS results (4). Sub-themes were identified using codes as shown in Table 1, Table 2, Table 3 and Table 4.

### 3.1. Dealing with the Urgency of the Metabolic NBS Result

The urgency of the metabolic NBS result was a common theme among GPs, encompassing discussions on the way of performing the communication, estimating the urgency of the specific IMD, and navigating the logistical hurdles they faced. Five out of ten GPs communicated the abnormal NBS result by performing a home visit (Table 1 (1a,1b)). Reasons for communicating the result via phone call, done by five out of the ten GPs, were lack of time and the GPs’ expectation of whether the family would understand the message by phone (Table 1 (1c)). Planning and logistics were a common struggle, and the urgency of the metabolic NBS result was often described as “creating challenges” in an already fully planned day (Table 1 (2a,2b,2c)). In addition, an estimation of the urgency of the abnormal NBS result had to be made to prioritize tasks (Table 1 (3a)).

**Table 1 IJNS-11-00062-t001:** Example Quotes: Theme 1.

Dealing with the Urgency of the Metabolic NBS Result—Quotes
**Sub-theme 1: Way of Performing the Conversation**
1a Participant 09: *“I went there* [to the parents’ home]. *Because that was her* [the pediatrician] *advice, I had to examine the child, whether it wasn’t lethargic for example.”* 1b Participant 12: *“I didn’t know whether they would be home, so I first made a phone call, told them that the new born screening result was abnormal, and then I performed a home visit.”* 1c Participant 08*: “I wanted to perform a home visit until I found out that she* [the mother] *was a GP herself… …The less highly educated people are, the sooner I would perform a home visit. But it also costs extra time, and you do need to have that extra time in that moment.”*
**Sub-theme 2: Logistical hurdles to perform the conversation**
2a Participant 07: *“…all of a sudden you are confronted with this result in a busy day, and they immediately expect you to do something with it* [the abnormal NBS result]… *…In a hectic day where you are already doing so many things at once, getting a result like this on your plate is pretty hard.”*
2b Participant 08: “*You are expected to immediately undertake action, all the while you also still have a completely different program running throughout the day… …I was in the middle of my outpatient clinic. So I felt urged to call the parents as quickly as possible with of course bad news. That felt very uncomfortable.”* 2c Participant 03: *“Ideally I would have performed a home visit. I think I would have done it* [a home visit] *if I wouldn’t have to pick up my child from daycare. If I would have had the time then I would have done it, but it just wasn’t possible.”*
**Sub-theme 3: Estimation of the Urgency**
3a Participant 01: “*I had no clue what kind of disease this was, and whether you had to undertake action immediately or not… Because that is also the question, do you have to see the child right now?… I think we were able to make a good decision on that, but that really was a decision made together* [with the pediatrician].”

### 3.2. The Role of the GP in the NBS Process

This theme emerged as GPs described their particular role in communicating the abnormal NBS result to parents. Where some GPs described their role as not to explain every detail of the IMD but being there for social support (Table 2 (1a,1b)), others expressed their discomfort and frustration with their role in the NBS process, leaving them feeling like messengers of unfamiliar information (Table 2 (1c,1d)). Participant 12 highlighted that communicating the abnormal NBS result adds to the already heavy workload. As GPs described their experience with communicating the NBS result, almost half indicated that they would tell parents that they have little knowledge about this particular IMD (Table 2 (2a,2b)).

Some GPs described their experience with performing the communication of the NBS result as quite positive, while another GP had a bad experience with performing the conversation due to wrong timing and bad logistics (Table 2 (3a–3c)). A factor that contributed to a positive experience with communicating the NBS result was the fact that the IMD pediatrician was included in the conversation by phone (on speaker) during the home visit (Table 2 (3a)).

**Table 2 IJNS-11-00062-t002:** Example Quotes: Theme 2.

The Role of the GP in the NBS Process—Quotes
**Sub-theme 1: The role of the GP**
1a Participant 01: *“I’m just there for emotional support. And I can of course examine the child, which the pediatrician is not able to do at that moment… …My role in this case is being there for the parents I would say.*
1b Participant 02: *“I could make a phone call for social support, but a clear explanation* [of the IMD] *is outside the scope of the GP.”*
1c Participant 08: *“As a GP you are not part of the newborn screening process. The test is taken by someone else, the results are interpreted by someone else, and then I am the one who all of a sudden needs to make a referral and needs to tell the parents. Well, that feels strange to me.”* 1d Participant 12: *“What I find strange is that we do not organize these newborn screening tests, the results that come from these tests, we often don’t have a clue of what they mean, it is so rare that you can read into it, but the next time you will have forgotten about it again. So you are actually giving parents someone else’s information, you are the messenger of something you don’t really know the answer to. (…) And that might be what’s behind all of it, a bit of frustration. And then it* [communicating the NBS result] *is also something that you actually don’t know anything about and ‘you just have to do it’. Well, I don’t like that very much.”*
**Sub-theme 2: Level of Experience with IMD**
2a Participant 01: *“…I just gave the message that something was wrong, we don’t exactly know how and to what degree. It is important that we are going to figure it out as soon as possible, and that we are going to call the pediatrician right now. …We should not expect the GP to give that specific information* [about the IMD].”
2b Participant 09: *“…That is also what I told the parents, that it really was my first time. But then you also feel a little bit of a nitwit, it feels a bit lame* [having little experience with the IMD], *and I can imagine that some parents say ‘well why are you the one coming to tell me then’, who actually are you, so to speak.”*
**Sub-theme 3: Satisfaction with Conversation**
3a Participant 01: [Did you miss something in the conversation? Would there be anything to improve?] *“I wouldn’t know. This* [abnormal NBS result] *gives such an incredible amount of stress about the health of your newborn, it is dramatic for parents, if you can relieve that even for a little by answering questions quickly, I think that worked out very well. If it is always arranged in this* [performing the conversation with the pediatrician on speaker] *way, then I think that would be beautiful.”* 3b Participant 06 (on paper): *“Not okay due to the wrong timing and bad logistics.”* 3c Participant 09: *“Well, I thought it* [the conversation with parents] *went very well, but I was quite nervous to have to go to the parents. I have been a GP for almost nine years, and I did not really know how this goes. So it took me by surprise.”* 3d Participant 12: [Do you feel like you were able to inform the parents sufficiently?] *“No, I actually don’t… Because of just being the messenger, and I don’t really think that that is my role as a GP.”*

### 3.3. The Current Organization of NBS in the Netherlands

A third theme focused on the way in which the communication of an abnormal NBS result is currently organized in the Netherlands, emphasizing the collaboration between the GP, the RIVM and the IMD pediatrician. Most GPs experienced the collaboration with both the medical advisor of the RIVM and pediatrician as overall pleasant (Table 3 (1a,1b)), although one GP indicated it felt more like passing the work off to him/her (Table 3 (1c)). The preliminary work (e.g., having called the pediatrician and having the right phone numbers for the GP) done by the medical advisor of the RIVM was described as being very helpful (Table 3 (2a)). While some GPs thought having the pediatrician on speaker during the conversation with parents would cost extra time, others thought it was of added value to be able to answer disease-specific questions right away (Table 3 (3a,3b)). In addition, an estimation of urgency made by the pediatrician can help in preparing the GP for the conversation (Table 3 (3c)). There were different opinions about whether the information provided by the medical advisor of the RIVM was sufficient to conduct the conversation. Some felt that knowledge about IMDs is by far too specialized for a GP and that the GP would not be able to answer questions about the IMD based on the information leaflet, while others considered the information in the leaflet enough to inform parents (Table 3 (4a–4c)).

**Table 3 IJNS-11-00062-t003:** Example Quotes: Theme 3.

The Current Organization of NBS in the Netherlands—Quotes
**Sub-theme 1: Collaboration with the Medical Advisor of the RIVM and Pediatrician**
1a Participant 03: *“I found that very pleasant, that you had very direct communication. I immediately received the right phone numbers and names, so that you don’t have to figure it all out yourself.”* 1b Participant 04 (on paper): [What did you think of the cooperation with the medical advisor of the RIVM and the pediatrician?] *“All very pleasant and skilled.”* 1c Participant 07: *“with the pediatrician fine, she supported me with what she could, I think. With the RIVM, it feels a little bit like passing work off to someone else, I don’t really know how to describe it, like here you have the phone call and the result and good luck.”*
**Sub-theme 2: Role of RIVM**
2a Participant 03: “*It was very nice that she had already taken action, the referral to the pediatrician was already arranged, an appointment with the pediatrician was already made, and she gave me the phone number of the pediatrician to discuss briefly before the conversation with parents. That was all very pleasant, that I didn’t have to enter the conversation completely blank. I, of course, had no idea what it* [the IMD] *was, so she* [the medical advisor of the RIVM] *already explained a lot and advised me to call the pediatrician for more explanation.”*
**Sub-theme 3: Role of Pediatrician**
3a Participant 12: [Did you consider calling the pediatrician during the conversation with parents?] *“The pediatrician suggested that, but I declined, saying that I simply don’t have the time or space for it.”*
3b Participant 01: *“I first called the pediatrician to discuss what I should pay attention to* [during the home visit]. *The pediatrician knew about the abnormal NBS result so could very well explain to me what I should look for, then I went there and agreed to call the pediatrician again during the conversation with parents. That went very well. …There are so many questions that I can’t answer during a conversation like this. …In this way* [with the pediatrician on speaker] *you can answer the questions and make a good first estimation of the situation.”* 3c Participant 03: *“The pediatrician gave me a bit of reassurance. I thought, a metabolic disease can be quite anxious, it could be very serious. But the pediatrician indicated that the metabolic disease that the girl possibly had was very good to live with, with for example a strict diet. That gave me some reassurance, and then you are able to enter the conversation with parents differently.”*
**Sub-theme 4: Information Provision**
4a Participant 04 (on paper): *“That was nice, since it isn’t about everyday medical information (for a GP), it is nice to have information and some explanation about the disease, but also about practical things like hospital appointments, what to take into account,* etc.*”* 4b Participant 07: [What did you think of the information provided in the disease-specific information leaflet?] *“Well it actually only included information about the specific patient and about what disease was found, no other medical content. It was very brief…. For now* [parents already had a child with the same IMD] *it was okay, but if it would have been the first time that I would have to confront parents with this news, it really would not have been enough for me. I would not be able to answer questions from parents sufficiently”* 4c Participant 12: [What did you think of the information provision? Was it sufficient?] *“Actually, no, I read something of which I have never heard, maybe during my study but that is far far away in my memory. No, for me it is new, it’s so specialized, we, as GPs, just don’t know that.”*

### 3.4. Evaluating Roles and Responsibilities in Communicating Abnormal Metabolic NBS Results

A variety of perspectives was expressed regarding alternative ideas for performing the communication of an abnormal metabolic NBS result. This theme includes the discussion about the involvement of different stakeholders: GPs, pediatricians, the RIVM and the Youth Healthcare Physician. One of the topics brought up in the discussion was the degree to which the GP should be involved during the NBS process. Most (*n* = 8) GPs suggested that due to the complexity of IMDs and the fact that the GP is generally not involved in the whole NBS process, the communication of an abnormal result to parents should be performed by the individuals that arrange the whole process (Table 4 (1a,1b)), while another GP felt that because of the familiarity with patients and parents, the GP is the right person to communicate the result (Table 4 (1c)).

Following the above opinions, suggestions were made about the most suitable person to perform the communication of an abnormal NBS result. Some (*n* = 2) GPs felt that the metabolic pediatrician would be best suited to perform the first communication, since they can answer all questions directly (Table 4 (2a,2b)). Others (*n* = 3) suggested that the Youth Healthcare Physician might be more suitable than the GP, since they are most in contact with the parents during the first few weeks of life (Table 4 (2c,2d)).

**Table 4 IJNS-11-00062-t004:** Example Quotes: Theme 4. *Child Health Clinic is a healthcare center for young children, offering regular check-ups and developmental support.

Evaluating Roles and Responsibilities in Communicating Abnormal Metabolic NBS Results—Quotes
**Sub-theme 1: Involvement of GP**
1a Participant 07: *“I also strongly question whether these kind of abnormal NBS results should be directed to us. I do understand that we are part of a specific branch within medicine with a broad range of responsibilities, but these* [IMDs] *are such niche topics that we encounter so rarely. I think even a general pediatrician would struggle to provide clear information about these cases, so you really need someone with specialized knowledge who can convey clear information to parents.”* 1b Participant 08: *“It would be best if the RIVM also contacts the parents, also because the pediatrician didn’t necessarily needed a referral. If it would be handled by the RIVM and we would stay out of it.”* 1c Participant 11: *“We thought about it before, what makes sense? We do not perform the newborn screening and also do not do the follow up. On the other hand, we do follow up the parents, and you do know the parents. So to me it makes sense that it is arranged this way, but the first time we came in touch with it* [communicating an abnormal NBS result]*, I thought, ‘hold on’, how does this work and does this make sense.”*
**Sub-theme 2: Shared Responsibility**
2a Participant 04 (on paper): *“I can imagine that in some situations the metabolic pediatrician directly contacts the parents. It could, for example, be an extra question when performing the newborn screening test: “Would you like to be contacted* via *phone in case of an abnormal NBS result, or would you rather be contact by your own GP about the abnormal NBS result?’. But again, as a GP, I want to be informed as soon as possible about this information!”* 2b Participant 07: *“I wonder that if the RIVM already had contact with the pediatrician, if they shouldn’t let the pediatrician contact the parents. Then a lot of questions can be answered straight away, they can give a lot more clarity about what is going on and what to expect.”* 2c Participant 09: *“I’m not really sure, but I do wonder whether it really has to be done by the GP. I think it might be more suitable for the youth health physician or the Public Health Service doctor. After all, the child is already well known at the Child Health Clinic * and the Public Health Service doctor. They just had the two-week screening. Meanwhile, they don’t typically come to me.”* 2d Participant 10: “*It is of course a public health issue, so you could also imagine the Public Health Service doctor taking it on. Perhaps that could provide a shorter line of communication. That said, there is a shortage of Public Health Physicians, so in that sense I’m not sure if it would be practical. But I do think it’s an interesting question, maybe it really is something for the Public Health Physician?”*

## 4. Discussion

To the best of our knowledge, this qualitative exploratory study is the first to describe the experiences with and opinions about communicating abnormal metabolic NBS results from a number of Dutch GPs’ points of view. Four key themes were identified: (1) dealing with the urgency of the metabolic NBS result, (2) the role of the GP in the NBS process, (3) the current standard in the Netherlands and (4) evaluating roles and responsibilities in communicating abnormal metabolic NBS results.

GPs are often called during an already fully planned day, requiring them to interrupt their work to address and communicate the abnormal metabolic NBS result. Half of the GPs interviewed in this study performed a home visit to communicate the abnormal metabolic NBS result, with lack of time being the main reason for not doing so. The way of performing the conversation (e.g., via phone, via home visit) plays an important role in parental satisfaction [6,13,14,15,16,17]. This was also observed in our previous study, where questionnaire results among 113 parents receiving an abnormal NBS result indicated that the main reason for satisfaction with the communication of an abnormal NBS result was a home visit performed by the GP [10]. Moreover, the urgency of the result differs per IMD, making it challenging for GPs to assess the need for an urgent referral. This may not only increase parental anxiety but could also lead to stress for the GP, as described by one participant.

Some GPs expressed that their primary role was to provide social support, rather than detailed disease-specific information about an IMD. Previous literature, mostly focusing on the abnormal NBS result for diseases such as cystic fibrosis and sickle cell disease, states that the content of information given to parents is shown to impact parents’ reactions [5,15,18,19]. This is supported by our previous study that demonstrated that the most important reason for unsatisfaction among parents receiving an abnormal NBS result was the receival of wrong information [10] parental anxiety [5,14,20]. Especially in diseases where confidence in treatment (like diet) from the early days is of the utmost importance, building trust through fair, transparent and comfortable communication is essential [21]. What added to a positive experience for the GP was having a phone call with the pediatrician before the conversation with parents and/or having the pediatrician on speaker during the conversation with parents, making sure disease-specific questions could be answered correctly right away. Not only the GPs’ satisfaction but also parental satisfaction is shown to increase when having the pediatrician on speaker during the first conversation as a ‘tandem conversation’ [10,16].

If we look at Europe and beyond, communicating an abnormal NBS result is performed by various healthcare providers and may be condition dependent. In the United Kingdom, for example, an abnormal metabolic NBS result is directly communicated to the pediatric metabolic team, who are then responsible for communicating the result within 24 h to the family [22] In Switzerland, receival of information is deliberately minimal until parents see the specialist where they receive accurate information [17,23]. Chudleigh et al. [23] showed that in the US, communication of an abnormal NBS result for CF is often done by the Primary Care Physician (comparable to GP in the Netherlands), whereas in Europe/Australia/New Zealand a range of professionals are responsible for the first communication [23]. Due to the limited availability of data on the type of healthcare provider performing the first communication of an abnormal metabolic NBS result, comparison between approaches and identification of the most effective approach remains difficult. When looking at guidelines for this first communication to parents in Europe, Burgard et al. [24] showed that in only 50% of the cases, communication to parents was regulated by guidelines. A lack of standardization raises concerns about inconsistencies in information delivery and ultimately parental satisfaction. Receival of accurate and up to date information during the first communication of the abnormal NBS result to parents is crucial and therefore raises the question of whether someone who only communicates such a result once or twice in their career is the most appropriate person to do so, especially without specific guidelines [23].

GPs in the Netherlands have been facing an increase in workload and administrative pressure in the last decade [25,26]. In addition, it can be questioned whether the GP is the right person to perform this initial communication to parents, acknowledging the expanding number of (very rare) diseases screened for in the NBS process. Although some GPs emphasized the value of their role in providing social support to parents, others suggested that direct communication between the metabolic pediatrician and the parents might be more effective. An additional suggestion was to involve the youth healthcare physician, as parents frequently interact with this service during the first year of the child for routine screenings and vaccinations, whereas the GP is typically only involved when a health issue arises. However, it should be taken into consideration that, according to Dutch health insurance policy, the pediatrician needs an official referral. In practice, pediatricians often tell the GP that a verbal referral suffices, and a written referral is not necessary, as also mentioned by two GPs. Incorporating alternative tools to help GPs in communicating the abnormal metabolic NBS result might be more feasible than shifting responsibilities. For instance, a disease-specific video for both GP and parents might aid in the explanation of the disease, while a “tandem conversation” with the metabolic pediatrician on speaker to directly address parental questions might help reduce anxiety. Such tools could offer more structured, accurate and reassuring communication to families receiving an abnormal NBS result.

There appears to be a mismatch between needs and expectations of parents regarding the communication and the support that GPs (can) offer. This discrepancy may be exacerbated by GPs’ limited disease-specific knowledge and insufficient time to prepare for the conversation, as highlighted in their quotes. Conversely, GPs emphasized the advantages of their role as a familiar figure to the family and their ability to examine the child when performing a home visit, both of which would not be possible if, for instance, a pediatrician communicated the results. Regardless of which healthcare provider communicates the abnormal NBS result, it should be acknowledged that proper communication and logistical preparation require time and cannot simply be added on top of already full schedules, particularly in cases involving language barriers, social difficulties or transportation issues. In Table 5, we explore alternative possibilities for the communication of an abnormal NBS result. Although some ideas might not be feasible (yet), they are included to provide a comprehensive overview of possible strategies and to foster further discussion. Experience/knowledge, availability and the possibility to perform a physical examination are difficult to combine in one healthcare professional, especially at short notice. This applies to all healthcare professionals mentioned in Table 5, suggesting new choices may have to be made based on what is considered the most important.

## 5. Limitations

The qualitative nature of this exploratory study gives potential for a more in-depth exploration relative to a survey, portraying the experiences and viewpoints of the interviewed GPs. However, qualitative data, just like quantitative data, remain open to subjective interpretation that may affect consistency in theme identification [27]. While recognizing that some bias will always exist, more than one coder per interview with discussion between coders to cross-check allows for mitigating bias as much as possible. In addition, if implemented, inter-coder reliability assessment can contribute to the systematicity and transparency of the coding process [28]. However, in accordance with Braun and Clarke’s approach to thematic analysis [11], no formal calculation of inter-coder reliability was performed in this exploratory study [11,29], while major differences in interpretations were discussed in the study team. Another limitation is that for some GPs, the conversation with parents took place up to one year ago, possibly leading to recall bias. There may have equally been some sampling and recruitment bias in terms of who decided to participate in the semi-structured interviews. Two GPs submitted their answers via email, potentially leading to loss of depth and misinterpretation due to the inability to ask follow-up questions. Lastly, the interviews were short i.e., 12–23 min. Further exploration and the use of more in-depth interview methods may have enhanced the quality of the data.

## 6. Conclusions

Together with our previous study [10], our aim was to portray both the parental perspective and the perspectives of a number of Dutch GPs towards the communication of an abnormal metabolic NBS result. The growing complexity of the NBS and the increasing workload, not only for Dutch GPs but possibly for all healthcare providers mentioned in Table 5, might suggest that alternative approaches or tools, such as a disease-specific video or a tandem conversation with the metabolic pediatrician, should be explored as potential solutions for delivering abnormal results even more effectively and accurately.

## Figures and Tables

**Table 5 IJNS-11-00062-t005:** Overview of present situation, and possibilities for the communication of abnormal NBS results.

Healthcare Provider	The Process of Communication and Referral	Considerations in Choosing Various Routes for Communicating Abnormal NBS Results
**General Practitioner ***	Calls parents or performs home visit **, informs them of positive NBS result, referral to metabolic pediatrician.	Strengths: Is a familiar face to the family. Possibility to perform home visit and examine the child. Limitations: Parents will not receive detailed information before reaching the metabolic pediatrician.
**Medical Advisor from RIVM**	Calls parents, informs them of positive NBS result, referral to metabolic pediatrician.	Strengths: Might be quicker than communication by the GP due to fewer logistical hurdles. Limitations: Not possible to examine the child. If parents cannot be reached by telephone or there is doubt on clinical severity, then another healthcare professional might still need to perform a home visit.
**Metabolic Pediatrician**	Calls parents, informs them of positive NBS result, invites them for diagnostic evaluation.	Might be feasible if the child is seen on the same day. Is being done in Switzerland in cases of a positive CF result [17]. Strengths: Parents will receive accurate information right away, might reduce parental anxiety. Limitations: Not possible to examine the child. If parents cannot be reached by telephone, then another healthcare professional might still need to perform a home visit.
**Youth Healthcare Physician**	Calls parents, might be able to perform home visit *, referral to metabolic pediatrician.	Strengths: Might be able to perform home visit and examine the child. Limitations: Is not used to see acutely ill patients. Parents will not receive detailed information before reaching the metabolic pediatrician.
**Specialized nurse**	Calls parents, informs them of positive NBS result, invites them for diagnostic evaluation.	Is currently being done in the UK for CF [23]. Strengths: Parents will receive accurate information right away. Might reduce parental anxiety. Limitations: Not possible to examine the child. If parents cannot be reached by telephone, then another healthcare professional might still need to perform a home visit.
**Midwife**	Can communicate result to parents during home visit *, referral to metabolic pediatrician.	Is being done in New Zealand, where midwives who are responsible for the care of the mother and child communicates the NBS result for CF [23]. Strengths: Is a familiar face to the family. Will be able to examine the child. Limitations: Parents will not receive detailed information before reaching the metabolic pediatrician.

* Current standard in the Netherlands. ** In case of a home visit, supplementary tools such as a disease-specific video or a tandem conversation with a metabolic pediatrician may support accurate information transfer.

## Data Availability

Data are available upon request.

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
