# Peer review of "Communication of an Abnormal Metabolic Newborn Screening Result in the Netherlands: A Qualitative Exploratory Study of the General Practitioner’s Perspective"

_2409-515X, 2025, doi:10.3390/ijns11030062_

Round 1
Reviewer 1 Report
Comments and Suggestions for Authors
Congratulations on this paper which highlights a number of key themes and advocacy points on behalf of the patients being identified with IMDs and their families.
I would ask for a correction about the UK procedure noted in the manuscript as the reference [21] used by authors is purely the approach for Cystic Fibrosis result communication, not IMDs. Whilst there is slight variation between england and the devolved nations (scotland, wales, northern ireland), for IMD communication of results, the key process is from newborn screening lab directly to the paediatric inherited metabolic team who are tasked with arranging follow-up with patient and family, ideally within 24hrs of the result being communicated by the newborn screening lab - so it is likely that it is a metabolic paediatrician do the communication of result +/- support from clinical nurse specialist. Authors may wish to review the pro/cons table 5 and associated narrative in the discussion to reflect this. I would signpost to https://www.gov.uk/government/publications/newborn-blood-spot-screening-laboratory-guide-for-imds/a-laboratory-guide-to-newborn-blood-spot-screening-for-inherited-metabolic-diseases#clinical-referral-and-follow-up--common-elements for the key reference for this. Alternatively, this could be referenced in the specialised nurse section to only be for CF testing, as has been acknowledged for the "Metabolic paediatrician" option from table 5 used in Switzerland. Similarly, it is worth noting again the approach for midwife is for CF, not wider IMD using reference [21].
Reviewer 2 Report
Comments and Suggestions for Authors
The paper addresses an interesting and relevant topic. However, its incremental value is minimal given that the same authors have previously published a parental perspective study on this subject. The conclusion is not unexpected: GPs' lack of IMD knowledge and their work is often characterized by time pressure. To significantly enhance its contribution, the authors should consider incorporating a deeper comparative analysis with their prior work or expanding on its policy relevance.
A significant concern lies with the study's methodology, particularly regarding its sampling and recruitment. The inclusion of only 12 General Practitioners (GPs), with two responses received via email, suggests potential sampling and recruitment biases that could impact the generalizability and robustness of the findings.
Finally, while the paper states that coding was performed by one author and reviewed by another, it lacks any discussion of inter-coder reliability. This omission is crucial, as it affects the trustworthiness and consistency of the qualitative data analysis.
N/A
